# Identification and Functional Analysis of Transcriptome Profiles, Long Non-Coding RNAs, Single-Nucleotide Polymorphisms, and Alternative Splicing from the Oocyte to the Preimplantation Stage of Sheep by Single-Cell RNA Sequencing

**DOI:** 10.3390/genes14061145

**Published:** 2023-05-25

**Authors:** Zijing Zhang, Qiaoting Shi, Xiaoting Zhu, Lei Jin, Limin Lang, Shijie Lyu, Xiaoling Xin, Yongzhen Huang, Xiang Yu, Zhiming Li, Sujuan Chen, Zhaoxue Xu, Wei Zhang, Eryao Wang

**Affiliations:** 1Institute of Animal Husbandry and Veterinary Science, Henan Academy of Agricultural Sciences, No. 116 Hua Yuan Road, Zhengzhou 450002, China; vincezhang163@163.com (Z.Z.); sqtsw@126.com (Q.S.); xtzhu2019@outlook.com (X.Z.); langlimin666@outlook.com (L.L.); sjlyu@outlook.com (S.L.); xuzhaoxue11@163.com (Z.X.); 2College of Animal Science and Technology, Northwest A&F University, Yangling 712100, China; hyzsci@nwafu.edu.cn; 3Henan Animal Health Supervision Institute, Zhengzhou 450003, China; 15838282287@163.com; 4Henan Provincial Animal Husbandry General Station, Zhengzhou 450008, China; hna1920@163.com; 5Synthetic Biology Engineering Lab of Henan Province, School of Sciences and Technology, Xinxiang Medical University, Xinxiang 453003, China; chensujuan101@163.com; 6College of Animal Science and Veterinary Medicine, Henan Institute of Science and Technology, Xinxiang 453003, China

**Keywords:** transcriptome profiles, long non-coding RNAs, single-nucleotide polymorphisms, alternative splicing, sheep, preimplantation, single-cell RNA sequencing

## Abstract

Numerous dynamic and complicated processes characterize development from the oocyte to the embryo. However, given the importance of functional transcriptome profiles, long non-coding RNAs, single-nucleotide polymorphisms, and alternative splicing during embryonic development, the effect that these features have on the blastomeres of 2-, 4-, 8-, 16-cell, and morula stages of development has not been studied. Here, we carried out experiments to identify and functionally analyze the transcriptome profiles, long non-coding RNAs, single-nucleotide polymorphisms (SNPs), and alternative splicing (AS) of cells from sheep from the oocyte to the blastocyst developmental stages. We found between the oocyte and zygote groups significantly down-regulated genes and the second-largest change in gene expression occurred between the 8- and 16-cell stages. We used various methods to construct a profile to characterize cellular and molecular features and systematically analyze the related GO and KEGG profile of cells of all stages from the oocyte to the blastocyst. This large-scale, single-cell atlas provides key cellular information and will likely assist clinical studies in improving preimplantation genetic diagnosis.

## 1. Introduction

From the development of the oocyte to the embryo, many dynamic and complicated processes take place as the zygote undergoes several rapid rounds of division and produces a mass of cells within the zona pellucida [1,2]. During these dynamic stages, differential gene expression in individual cells is a key determinant of cellular differentiation, function, and physiology [3,4]. In recent years, several studies have documented the key developmental processes underlying the formation of blastomeres of 4- and 8-cell embryos of mice [5], cattle [6], and goats [7]. However, the exact mechanism and developmental patterns underlying how the blastomeres of the 2-, 4-, 8-, 16-cell, morula, and blastocyst stages undergo asymmetric division are still unclear. Understanding the molecular mechanism underlying cleavage-stage development is critically important for improving preimplantation genetic diagnosis.

Single-cell RNA sequencing (scRNA-seq) provides an alternative method for studying the cellular heterogeneity of human preimplantation embryos and embryonic stem cells [8,9], mouse oocytes [10], porcine oocyte maturation [11], and Haimen white goat’s oocyte maturation by generating a readout of the abundance of a transcript within a cell [12]. Indeed, scRNA-seq applied to mammalian gametes has generated new insights into the composition of transcripts and disease-related functional abnormalities. However, scRNA-seq studies have not thoroughly characterized how the blastomeres of the 2-, 4-, 8-, 16-cell, and especially the morula and blastocyst stages, undergo asymmetric division. Here, we aimed to transcriptionally profile nucleated cells present during the blastomeres of 2-, 4-, 8-, 16-cell and morula stage undergoing asymmetric division to provide a broad profile of the composition of transcripts in the cell and their transcriptomes.

Long non-coding RNAs (lncRNAs) are a diverse group of RNAs that are longer than an arbitrary limit of 200 nt and do not encode proteins [13]. Nevertheless, lnc RNAs can be located in exonic, intronic, and intergenic regions and can regulate gene expression by interacting with other biological macromolecules, such as RNA, DNA, proteins, and other factors, to promote normal cell function [14,15]. Compared with the characteristics of protein-coding genes, lncRNAs tend to be less conserved across species and often show lower expression levels and high tissue specificity [16]. During the development of embryos, some lncRNAs, such as Xist, Tsix, and H19, have significant regulatory functions and can potentially determine the cell’s fate and direction of differentiation during embryogenesis to form different organs or special tissues that contain various cells expressing specific genes [17,18,19].

An increasing number of studies has reported several thousands of annotated lncRNA loci in mammalian oocytes and early embryos [1,20]. For example, a total of 2733 novel lncRNAs was found to be expressed in specific developmental stages among 8701 lncRNAs using single-cell sequencing analysis of 124 individual cells from human embryonic stem cell (ES) and human preimplantation embryos at different passages. In addition, 5204 novel lncRNAs were obtained from in vivo and somatic cell nuclear transfer mouse preimplantation embryo, suggesting that several lncRNAs and their association with known protein-coding genes might be involved in mouse embryonic development and cell reprogramming [8,21]. Another study reported that approximately a quarter of the 1600 lncRNA loci expressed during the murine oocyte-to-embryo transition employed promoters from a long terminal repeat retrotransposon class, which exhibited either maternal or zygotic expression and showed signatures of massive expansion in the evolution of the rodent genome [16]. In bovine early embryos, some lncRNAs play a role in the translational control of target mRNA and are thus important for managing maternal reserves, especially before embryonic genome activation, as these reserves contain the embryonic program [22]. Despite the fact that various functional lncRNAs are important during embryonic development, the effect of lncRNAs on the blastomeres of the 2-, 4-, 8-, 16-cell, and morula stages of development has not been studied; nevertheless, this subject requires more attention and discussion by scientists. LncRNAs play an important role in biological processes, including epigenetic regulation, dosage compensation, cell cycle, cell differentiation, proliferation, apoptosis through gene imprinting, chromatin remodeling, transcriptional activation, transcriptional interference, and nuclear splicing regulation.

Here, we conducted an scRNA-seq survey of 24 sheep cells during their development from the oocyte to the blastocyst stages. We then conducted experiments involved in the identification and functional analysis of transcriptome profiles, lncRNAs, single-nucleotide polymorphisms (SNPs), and alternative splicing (AS). Using these different methods, we constructed a profile to characterize cellular and molecular features and systematically analyze the related GO (Gene Ontology) and KEGG (Kyoto Encyclopedia of Genes and Genomes) profile of cells during sheep development. This large-scale single-cell atlas provides key cellular information and will likely aid clinical studies in the development of more efficient preimplantation genetic diagnosis.

## 2. Materials and Methods

### 2.1. Animals and Sample Collection

All work with animals was completed in accordance and with the approval of the Henan Academy of Agricultural Sciences institutional animal care and use committee. Mature sheep were obtained from Hui yuan Sheep Industry Co., Ltd. (Puyang, Henan province, China). We used 15-month-old female sheep (40 kg) for our study. The animals were provided with grass and drinking water and had access to an animal exercise pen. All animals were healthy, showed a normal appetite, and had smooth wool. Artificial insemination using semen from one of three fertile rams was conducted at 12- and 24-h post-standing heat (Day 0). Donor animals were anesthetized and in vivo developed oocytes and embryos at the 2- to 16-cell stages were collected by oviductal flushing 2–4 days after estrus. Early morulae, compact morulae, and blastocysts were collected by routine non-surgical uterine flushing on days 5, 6, and 7. All oocytes and embryos were examined and staged under light microscopy. Only morphologically intact embryos meeting the standards of Grade 1 by the International Embryo Transfer Society were included in this study. Embryos were washed twice in D-PBS before being frozen and stored individually in RNAlater (Ambion, Grand Island, NY, USA) in liquid nitrogen.

### 2.2. RNA Isolation, Library Preparation, and Sequencing 

First, total RNA was isolated from the sheep sample (three biological replicates per sample combined) using single cell RNA kit (Single Cell RNA Purification Kit NGB) and extracted (Norgen Biotek, CA, USA) following the manufacturer’s procedure. The single cell RNA quantity and purity were analyzed with the Bioanalyzer 2100 and RNA 6000 Nano LabChip Kit (Agilent, CA, USA) with RIN number > 7.0. Approximately 10 μg of total RNA representing a specific adipose type was subjected to isolate Poly (A) mRNA with poly-T oligo-attached magnetic beads (Invitrogen, Carlsbad, CA, USA). Following purification, all amplifications were carried out in parallel with positive and no-template controls for quality assurance using the SMARTer Universal low Input RNA kit (Clontech) for cDNA library. Briefly, RNA was converted to cDNA, and then adapters for Illumina sequencing (with specific barcodes) were added through PCR using only a limited number of cycles. The PCR products were purified (Protocol C), and then ribosomal cDNA was depleted. The cDNA fragments were further amplified with primers universal to all libraries. Lastly, the PCR products were purified once more to yield the final cDNA library. Then, the mRNA was fragmented into small pieces using divalent cations under elevated temperature. Then the cleaved RNA fragments were reverse-transcribed to create the final cDNA library in accordance with the protocol for the mRNASeqsample preparation kit (Illumina, San Diego, CA, USA); the average insert size for the paired-end-libraries was 300 bp (±50 bp). Following this, we performed the paired-end sequencing on an IlluminaHiseq4000 at the (LC Sciences, Houston, TX, USA) using the vendor’s recommended protocol.

### 2.3. Quality Control and Assembly of Transcriptome Data 

Raw data in FASTQ format were first processed through in-house perl scripts. Clean reads were obtained by removing reads containing adapters, reads containing poly-N, and low-quality raw reads. Furthermore, Q20, Q30, and GC contents of the clean data were calculated. All of the downstream sequencing analyses were based on high-quality clean reads. For the RNA-seq data, all clean reads from each sample were aligned to the sheep reference genome (https://www.ncbi.nlm.nih.gov/genome/83?genome_assembly_id=351950, accessed on 1 October 2019) using Tophat v2.0.949. The distribution of known gene types was analyzed by HTSeq software. The mapped reads of each sample were then assembled by both Scripture (beta2) and Cufflinks (v2.1.1) using a reference-based approach. 

### 2.4. RNA-Seq Reads Mapping

We aligned reads to the UCSC (http://genome.ucsc.edu/, accessed on 1 October 2019) sheep reference genome using the Tophat package, which initially removes a portion of the reads based on quality information accompanying each read and then maps the reads to the reference genome. Tophat allows multiple alignments to be read (up to 20 by default) and a maximum of two mismatches when mapping the reads to the reference. Tophat builds a database of potential splice junctions and confirms these by comparing the previously unmapped reads against the database of putative junctions.

### 2.5. Estimation of Transcript Abundance and Differential Expression and Principal Component Analysis (PCA)

The mapped reads of each sample were assembled using StringTie. All transcriptomes from the samples were then merged to reconstruct a comprehensive transcriptome using perl scripts. After the final transcriptome was generated, StringTie and Ballgown were used to estimate the expression levels of all transcripts. StringTie was used to determine the expression level of mRNAs by calculating FPKM (fragments per kilobase million). Differentially expressed mRNAs and genes were identified if log2 (fold change) > 1 or log2 (fold change) < −1 and by statistical significance (*p*-value < 0.05) through the R package (1.12.0) Ballgown and then used to generate a PCA plot, showing the relationship of gene expression between the different stages as a previous study [9].

### 2.6. WGCNA Analysis 

The co-expression network analysis was performed using WGCNA (version:1.61) [23]. First, the soft threshold for network construction was selected. The soft threshold constrains the adjacency matrix to assume a continuous value between 0 and 1 so that the constructed network conforms to the power-law distribution and is closer to the real biological network state. Second, the scale-free network was constructed using the blockw iseModules function, followed by the module partition analysis to identify gene co-expression modules, which groups genes with similar patterns of expression. The modules were defined by cutting the clustering tree into branches using a dynamic tree-cutting algorithm and assigning different colors for visualization to the modules [23]. The module eigengene (ME) of each module was then calculated. ME represents the expression level for each module. Next, the correlation between ME and the clinical trait in each module was calculated, followed by the determination of gene significance.

### 2.7. Transcript Assembly and Identification of Candidate lncRNAs 

First, Cutadapt was used to remove the reads that were contaminated with adaptors [24], low-quality bases, and undetermined bases. Sequence quality was then verified using FastQC (http://www.bioinformatics.babraham.ac.uk/projects/fastqc/, accessed on 1 October 2019). We used Bowtie2 (FastQC) and Tophat2 to map read to the genome [25]. The mapped reads of each sample were assembled using StringTie [26]. All transcriptomes from the samples were then merged to reconstruct a comprehensive transcriptome using perl scripts. After the transcriptome was generated, StringTie and Ballgown were used to estimate the expression levels of all transcripts. Transcripts that overlapped with known mRNAs and transcripts shorter than 200 bp were discarded. We utilized CPC and CNCI to predict transcripts with coding potential [27]. All transcripts with CPC scores < −1 and CNCI scores < 0 were removed. The remaining transcripts were considered lncRNAs.

### 2.8. Classification of lncRNAs

The annotated lncRNAs were subdivided into the following four categories based on their locations relative to the nearest protein-coding genes: (i) lncRNAs that do not overlap protein-coding genes (lincRNAs); (ii) lncRNAs located entirely within a protein-coding locus (intragenic lncRNAs); (iii) lncRNAs partially overlapping a protein-coding gene (overlapping lncRNAs); (iv) lncRNAs that overlapped exons of a protein-coding transcript on the opposite strand (antisense lncRNAs). Perl scripts were developed to classify these four categories. 

### 2.9. Quantification and Differential Expression Analysis 

The relative abundances of both candidate lncRNAs and coding genes in each sample were computed by calculating the FPKM using Cufflinks (v2.1.1). Differentially expressed lncRNAs in comparison groups were identified using the Cuffdiff program. For biological replicates, transcripts with adjusted *p*-values < 0.05 were considered differentially expressed between the two groups. 

### 2.10. Predictions of Cis and Trans-Target Genes 

To explore the function of lncRNAs, we first predicted the cis and trans-target genes of lncRNAs. We searched for coding, cis-target genes 10 k and 100 k upstream and downstream, respectively, of candidate lncRNAs, and then analyzed their functions. We calculated the expressed correlation between lncRNAs and coding genes with custom scripts and then analyzed their functions through functional enrichment analysis. The trans-target genes and lncRNAs were identified by their expression levels. 

### 2.11. SNP Analysis

To further characterize the SNPs, we categorized them as genic or intergenic. Approximately half of the SNPs (47%) were located in genic regions, and the rest was located in intergenic regions. In addition, nonsynonymous SNPs in the exon region were analyzed to determine whether their amino acid character had changed (e.g., hydrophobic to basic or stop codons), given that compositional changes of the amino acids in proteins can result in changes in structural conformation or enzymatic activities and thus generate phenotypic diversity or critical functional variations. First, 20 amino acids were clustered into several character groups. Non-synonymous SNPs that caused amino acid changes from one group to another were searched. Common SNPs in the eight groups representing each line were then classified.

### 2.12. AS Data Collection

Data on 96 melanoma cases with clinicopathological information were obtained to explore changes in AS events. To analyze the AS profiles for each patient, the SpliceSeq tool (version 2.1), a java application, was used to evaluate the splicing patterns of mRNA in the melanoma cohort. The percentage spliced in value was calculated to quantify alternative splicing events and ranged between 0 and 1 for seven types of AS events, including exon skip (ES), alternate terminator (AT), and mutually exclusive (ME).

### 2.13. GO and KEGG Enrichment Analysis 

GO enrichment analysis of the aforementioned groups was conducted using the GOseq R package while correcting for biases in gene length. In addition, KOBAS software and the KEGG database (http://www.genome.jp/kegg/, accessed on 1 October 2019) were used to analyze the statistical enrichment of target genes of differentially expressed lncRNAs in KEGG pathways. Lower *p*-values corresponded to more relevant pathways, and corrected *p*-values < 0.05 were considered significantly enriched by DEGs.

### 2.14. Statistical Analysis

Statistical analysis was performed using SPSS13.0 software. Proportional data were compared using chi-squared analysis or Fisher’s exact tests, and differences were considered significant when *p* < 0.05. The percentage of blastocyst formed represented the number of blastocysts formed divided by the total number of embryos cultured. The percentage of high-quality blastocyst formed represented the number of high-quality blastocysts divided by the total number of blastocysts cultured.

## 3. Results

### 3.1. Transcriptome Profiles

To establish single-cell transcriptome profiles during the blastomeres of the 2-, 4-, 8-, 16-cell, and morula stage as they underwent asymmetric division, we used previously published protocols; single cells were subjected to RNA-seq sample preparation with several steps modified [8]. Overall, scRNA-seq was performed on 24 cells samples using the Illumina HiSeq2000 sequencer (Table 1). We generated 384 Gb of sequencing data from the 24 cells samples, with, on average, 10.2 million reads per cell with read lengths of 100 bp. To determine whether these gene expression profiles correlated with developmental stages, we analyzed RNA-seq data from blastocyst cells of the oocytes using a heat map. The greatest changes in gene expression were observed between the morula and blastocyst stages, which may be explained by the major morula–blastocyst transition and are largely representative of the expression patterns occurring during sheep preimplantation development (Figure 1A). Weighted gene co-expression network analysis (WGCNA) is commonly used to reveal correlations between genes in different samples. Previous WGCNAs have been used to reveal developmentally important genes of the bovine sham nuclear-transfer (shNT) blastocysts [28]. Following the approach of a previous bioinformatics study [23], the soft threshold for network construction was set to 30 (Figure 1B). Meanwhile, the fitting degree of the scale-free topological model was 0.85. Thus, this network conformed to the power-law distribution and was closer to the real biological network state. A total of 26 modules (Figure 1C) was obtained in the current study. The differentially expressed genes (DEGs) in grey were not included in any module; thus, subsequent analyses were no longer performed on DEGs in grey. Mutually exclusive (ME) was in accordance with the expression pattern of DEGs in each module. The pattern of similar within-stage and different between-stage expression patterns were also supported by principal component analysis (PCA) (Figure 1D). Notably, differences in the expression patterns between oocytes, 4-cell, 8-cell, and blastocyst stages confirmed the key timing of the transcriptome profiles throughout the various stages of sheep development.

### 3.2. Differentially Expressed Genes

Although the total numbers of genes expressed in different stages from the oocyte to the blastocyst varied little, the identities of the genes expressed during early development were dramatically different (Table 2). A total of 2127 unique genes were identified to be differentially expressed among all of the developmental stages (*p* < 0.05). Consistent with the Pearson correlations between all detected genes, the majority of the DEGs (1948) were identified between the morula and blastocyst cell stages, indicating that the greatest changes in gene expression occurred during this transition. Among these genes, 1092 and 856 were down- and up-regulated, respectively (Figure 2). For example, between the oocyte and zygote groups, significantly down-regulated genes included *BTF3* (basic transcription factor 3), *TLR1* (toll-like receptor 1), and *SPINT2* (serine peptidase inhibitor, Kunitz type 2), while significantly up-regulated genes included *PEX12* (peroxisomal biogenesis factor 12) and *PGK1* (phosphoglycerate kinase 1). The second-largest change in gene expression occurred between the 8- and 16-cell stages. A total of 1211 DEGs were detected, and 724 and 487 were down- and up-regulated, respectively (Figure 2).

### 3.3. Gene Ontology and Kyoto Encyclopedia of Genes and Genomes Analyses of DEGs

Between the oocytes and the zygote, 61 and 207 of the 268 DEGs were down- and up-regulated, respectively. These genes represent a rapid degradation of the maternally stored transcripts. Gene ontology (GO) analysis indicated that there was a significant over-representation of elements involved in nuclear speck and cytoplasm. The Kyoto Encyclopedia of Genes and Genomes (KEGG) analysis showed that most DEGs were primarily involved in the spliceosome, ribosome biogenesis in eukaryotes, and Epstein–Barr virus infection pathway Appendix A, Figure 3A(1,2). Between the zygote and the 2-cell stage, 266 and 232 of the 498 DEGs were down- and up-regulated, respectively. GO analysis indicated that there was a significant over-representation of elements involved in translation, in the structural constituent of ribosome, inRNA binding, and in extracellular exosome. The KEGG analysis showed that most DEGs were primarily involved in RNA transport and the ribosome pathway (Appendix A, Figure 3B(1,2)). Between the 2-cell and the 4-cell stage, 58 and 40 of the 98 DEGs were down and up-regulated, respectively. GO analysis indicated that there was a significant over-representation of elements involved in the positive regulation of cell proliferation, negative regulation of transcription, DNA-templated, and negative regulation of cell differentiation. The KEGG analysis showed that most DEGs were primarily involved in the phagosome pathway (Appendix A, Figure 3C(1,2)). Between the 4-cell and the 8-cell stage, 146 and 80 of the 226 DEGs were down- and up-regulated, respectively, which was inconsistent with the findings of a previous study that used scRNA-seq to profile human preimplantation embryos and embryonic stem cells [23,29]. GO analysis indicated that there was a significant over-representation of elements involved in the nucleus. The KEGG analysis showed that most DEGs were primarily involved in the cell cycle pathway (Appendix A, Figure 3D(1,2)). Between the 8-cell and the 16-cell stage, 487 and 724 of the 1211 DEGs were down- and up-regulated, respectively. GO analysis indicated that there was a significant over-representation of elements involved in RNA binding and mitochondrion. The KEGG analysis showed that most of the DEGs were primarily involved in the ribosome pathway (Appendix A, Figure 3E(1,2)). Between the 16-cell and the morula stage, 97 and 67 of the 174 DEGs were down and up-regulated, respectively. GO analysis indicated that there was a significant over-representation of elements involved in transferase activity. The KEGG analysis showed that most of the DEGs were primarily involved in the DNA replication pathway (Appendix A, Figure 3F(1,2)). The majority of the DEGs, 856 and 1092 of the 1948 DEGs were down- and up-regulated, respectively, between the morula and blastocyst stages. GO analysis indicated that there was a significant over-representation of elements involved in extracellular exosome and cytosol. The KEGG analysis showed that most of the DEGs were primarily involved in the transcriptional misregulation in cancer, lysosome, and protein processing in the endoplasmic reticulum pathway (Appendix A, Figure 3G(1,2)). 

### 3.4. Genome-Wide Discovery and Identification of lncRNAs

To identify new lncRNAs involved in sheep development from the oocyte to the blastocyst stages, cell samples from oocytes, zygote, and blastomeres of the 2-, 4-, 8-, 16-cell, morula, and blastocyst stages were collected. Transcriptome sequencing was then performed using the Illumina HiSeqTM 4000 platform. An overview of the analysis pipeline is shown in Figure 4. After removing adaptor reads, reads containing poly-N > 10%, and low-quality reads, clean reads were obtained for each sample and used in the following analyses. The GC content of each sample was between 43.5 and 45%. Thus, approximately 75.28–91.45% of the total clean reads could be mapped to the mouse reference genome sequence using Tophat v2.0.9 (Table 3). The different gene subtypes of the above-mapped reads are shown in Figure 1B and are based on genomic overlap with existing annotations using the HTseq program. A total of 274,470 assembled transcripts were produced using both Scripture (beta2) and Cufflinks (v2.1.1). 

### 3.5. Features of lncRNAs

The fragments per kilobase of exon per million fragments mapped (FPKM) values and numbers of lncRNAs demonstrated that lncRNAs in cell samples from the oocyte, fertilized egg, 2-, 4-, 8-, 16-cell, morula, and blastocyst stages were expressed at lower levels compared with the levels of protein-coding genes expressed (Figure 5A). However, the lncRNA transcript length was mostly ≥1000 bp, which was not significantly different compared with transcript lengths observed for protein-coding genes (Figure 5B). In addition, significant differences in the distributions of exon number between protein-coding genes and lncRNAs were also detected, and 82.41% of all lncRNAs only contained two exons (Figure 5C). Furthermore, most of the lncRNAs contained relatively shorter open reading frames (ORFs) compared with protein-coding genes (Figure 5D).

### 3.6. Differentially Expressed lncRNAs

A total of 42 differentially expressed lncRNAs (*p* < 0.05; 52 transcript isoforms) were detected (Figure 6). The most noticeable changes in lncRNA expression occurred between the morula and blastocyst stages, in which 19 (23 transcript isoforms) were significantly up-regulated and six (10 transcript isoforms) were down-regulated (Figure 6). Overall, differentially expressed lncRNA transcripts were fewer in number in sheep compared with the implantation and inter-implantation sites of pregnant mice [30].

### 3.7. Cis- and Trans-Target Genes of lncRNAs

To investigate the function of lncRNAs, we first predicted the putative lncRNA cis- and trans-regulatory target genes. Protein-coding genes located within 100 kb upstream and downstream of lncRNAs were considered cis-targets (Appendix A). The trans-target genes located far from lncRNAs are shown in Appendix A.

### 3.8. Functional Analysis of Differentially Expressed lncRNAs

To predict the function of lncRNAs during the different aforementioned stages, GO and KEGG analyses of the cis and trans-target genes of the lncRNAs in the eight comparison groups were performed. GO analysis of the cis-targets revealed only one significantly enriched GO term (mitochondrion) in the oocyte vs. zygote stages. The KEGG analysis revealed that the significantly enriched pathways in the oocyte vs. zygote stages were “Spliceosome” and “Carbon metabolism” (*p* < 0.05, Figure 7A(1,2) and Appendix A). The significantly enriched GO terms of the cis-targets in the zygote vs. 2-cell stages, which represented biological processes and molecular functions, were associated primarily with nucleus, mitochondrion, extracellular exosome, cytosol, and cytoplasm. The KEGG analysis revealed that the significantly enriched cis pathways in the oocyte vs. zygote stages were “Non-alcoholic fatty liver disease,” “Huntington’s disease,” and “Alzheimer’s disease” (corrected *p*-value < 0.05, Figure 7B(1,2) and Appendix A). There was only one significantly enriched GO term detected in the 2-cell vs. 4-cell stages: intracellular signal transduction. The KEGG analysis revealed that the significantly enriched cis pathways were the “MAPK signaling pathway” and “Glycerophospholipid metabolism” (*p* < 0.05, Figure 7C and Appendix A). Furthermore, no GO terms were significantly enriched in the 4-cell vs. 8-cell stages. The KEGG analysis revealed that the only significantly enriched cis pathway was “cell cycle” (corrected *p*-value < 0.05, Figure 7C(1,2) and Appendix A). In the 8-cell vs. 16-cell stages, there were additional significantly enriched GO terms: nucleus, membrane, and others. The KEGG analysis revealed that the only significantly enriched cis pathway was “Ribosome” (corrected *p*-value < 0.05, Figure 7E(1,2) and Appendix A). Seven significantly enriched GO terms were detected in the 16-cell vs. morula stages: nucleotide binding, membrane, and others. The KEGG analysis revealed that the significantly enriched cis pathways were “RNA transport” and “AMPK signaling pathway” (*p* < 0.05, Figure 7F(1,2) and Appendix A). In the morula vs. blastula stages, there were additional significantly enriched GO terms: RNA binding, nucleus, and others. The KEGG analysis revealed that the significantly enriched cis pathways were “Huntington’s disease” and “Oxidative phosphorylation” (corrected *p*-value < 0.05, Figure 7G(1,2) and Appendix A). 

### 3.9. Distribution of Different SNP and Indel Types in Sheep from Oocyte to Blastocyst Development

Single-nucleotide polymorphisms (SNPs) are the most common form of genetic variation in humans and drive phenotypic variation. Due to evolutionary conservation, SNPs and indels (insertion and deletions) are depleted in functionally important sequence elements [31,32]. Here, using the SAMtools/Popoolation software package. A total of 4,352,847 putative SNPs and 297,411 INDEL was predicted, with an average of 181,368 SNPs and 12,392 INDEL per sample, respectively. After removing redundant SNPs among all samples, we had 678,433 and 8454 INDEL from SAMtools/Popoolation2 (Appendix A). Then, concerning all the putative SNPs in sheep from oocyte to pre-implantation development, there were 79,516, 70,858, 99,129, 115,926, 72,615, 78,782, 66,420, 224,557 intergenic SNPs. In addition, 47,852, 40,178, 60,970, 67,126, 37,762, 64,925, 39,054, 201,642 SNPs from different stages in sheep from oocyte to pre-implantation development were genic, and defined exactly as in the 5′UTR, 3′UTR, and upstream and downstream of protein-coding genes. Furthermore, in these three SNP datasets, there were large percentages of intergenic (including upstream/downstream) SNPs (37–49%). There were 79,516, 70,858, 99,129, 115,926, 72,615, 78,782, 66,420, 22,4557 intergenic SNPs. In addition, 47,852, 40,178, 60,970, 67,126, 37,762, 64,925, 39,054, 201,642 SNPs from different stages in sheep from oocyte to pre-implantation development were genic, and of these genic SNPs (Table 4).

### 3.10. SNP and Indel Functional Annotation

Functional annotation of SNPs with allelic imbalances was performed using the Blast2GO suite [32,33,34,35]. The SNP-flanking sequences were searched against the NCBI nr-protein database using BLASTx. Associated genes and GO terms were then obtained. In the biological processes’ category, SNP genes were associated with various cellular processes that were primarily involved in development-related mechanisms, including the regulation of the MyD88-dependent toll-like receptor signaling pathway, regulation of metabolic and oxidation-reduction processes, and protein translation. In the molecular function category, SNP-containing genes were associated with binding phosphoprotein, nucleic acid, and actin. In addition, a significant number of the genes were associated with transferase, motor, oxidoreductase, and structural molecule activities. In the cellular component category, many of the genes were associated with the cytoplasmic compartment, membranes, myosin complex, and extracellular region compartment (Appendix A), which is consistent with the findings of previous studies [31].

Additionally, KEGG pathway mapping was used to assign functions to the SNP-containing transcripts. A search of transcripts against the KEGG database yielded 1043 transcripts (13.15%) with significant hits to 632 KEGG Orthologies belonging to different pathways (Appendix A).

### 3.11. Overview of AS Events in Sheep from Oocyte to Blastocyst Development

Splicing events were comprehensively analyzed for 24 single cells based on relevant RNA-seq data. In the oocyte stage, there were in total 101,835 AS events detected in 54,266 genes, comprising 7003 alternative exon ends (AE) events detected in 2603 genes, 1862 retention of single (IR) events in 723 genes, 146 multiple (MIR) intron events in 56 genes, 5260 multi-exon SKIP (MSKIP) events in 1400 genes, 17,236 Skipped exon (SKIP) events in 4310 genes, 33,409 Alternative 5′ first exon (TSS) events in 20,899 genes, 26,563 alternative transcription termination site (TTS) events in 20,900 genes, 2063 approximate AE (XAE) events in 729 genes, 1055 approximate IR (XIR) events in 359 genes, 76 approximate MIR (XMIR) events in 31 genes, 1969 approximate MSKIP (XMSKIP) events in 631 genes, and 5193 approximate SKIP (XSKIP) events in 1625 genes (Figure 8A). Next, splicing events were comprehensively analyzed in fertilized egg stage, 2-cell, 4-cell, 8 cell, 16 cell, morula and blastocyst shown in Figure 8B–H. Of those stags, in morula and blastula, only one type of AS event was detected in most genes, although there were some exceptions; generally, it was demonstrated that three or more splicing events could be attributed to one gene, with a maximum of five types of AS events observable for a single gene. However, TSS was the predominant type of event in all the histologic STS subtypes, which revealed that TSS was the most common splicing event in sheep from oocyte to blastocyst development.

### 3.12. Associated AS Events

The main AS events were SKIP, TSS, and TTS (Figure 1). Therefore, we analyzed associated AS events and genes by UpSet plot. SKIP was detected in 2566 genes at every stage (Figure 9A). Furthermore, TSS and TTS were detected in 11,441 genes at every stage (Figure 9B,C). More than one AS event could occur in one gene, and up to three types of splicing events were identified in one gene. Thus, one gene might have two or more AS events that were markedly related to the PFI of PTC patients. The UpSet plot revealed that TSS was the most common prognosis-related event.

### 3.13. Molecular Characteristics of the Most Important AS Events

Based on UpSet plot, there were 13,663 genes with one or more AS events at different stages (Figure 10A). We then carried out GO and KEGG analysis. The functional annotations revealed that “regulation of transcription, DNA templated (884 genes),” “transport (609 genes),” and “proteolysis (397 genes)” were the three most significant biological process terms. “Membrane (4323 genes),” “nucleus (3966 genes),” and “integral component of membrane (3055 genes)” were the three most significant cellular component terms. For molecular function, “ATP binding (1498 genes),” “nucleotide binding (1441 genes),” and “nucleic acid binding (1412 genes)” were the three most enriched categories (Figure 10B). 

Furthermore, we found that the related pathways were metabolism, environmental information processing, and human diseases. The “metabolism” and human diseases correlated pathways were mostly genes involved in purine metabolism (131 genes), pyrimidine metabolism (93 genes), and pathways in cancer (181 genes) (Figure 10C). In our study, there were 13,663 genes with one more AS event suggesting that AS may play an important role at different developmental stages. Furthermore, “ATP binding (1498 genes),” “nucleotide binding (1441 genes),” and “nucleic acid binding (1412 genes)” were the three most enriched categories, suggesting that alternative protein isoforms with distinct functions are expressed. Thus, defects in splicing control might be able to induce losses in function and severe phenotypes and require further study.

## 4. Discussion

In recent years, much research has focused on the study of the development of blastomeres of the 4- and 8-cell embryos of mice [5], cattle [6], and goats [7]. However, the exact mechanism and the developmental patterns underlying how the blastomeres of the 2-, 4-, 8-, 16-cell, morula, and blastocyst stages undergo asymmetric division are still unclear. For the first time, we used scRNA-seq to study the transcriptome profiles during sheep development from the oocyte to the blastocyst stages. Our data showed that from the 4-cell to the 8-cell stage, there were no noticeable changes in the transcriptome profile as has been shown for the 4- and 8-cell embryos of mice [6,7,36] (Figure 1A). However, the first major change was noted between the 8-cell and 16-cell stages, which is similar to the pattern observed in the bovine embryonic genome. The greatest changes in gene expression were observed between the morula and blastocyst stages, which may be explained by a major morula–blastocyst transition that results in expression patterns characteristic of sheep preimplantation development [11,37]. Furthermore, 1092 and 856 DEGs were down- and up-regulated, respectively (Figure 2). *BTF3* (basic transcription factor 3), *TLR1*(toll-like receptor 1), and *SPINT2* (serine peptidase inhibitor, Kunitz type 2) were markedly up-regulated, while *PEX12* (peroxisomal biogenesis factor 12) and *PGK1* (phosphoglycerate kinase 1) were significantly down-regulated between the oocyte and zygote stages. In others reports, BTF3 as one of the important transcription factors was proved in gastric cancer development and in progression by enhance transcription [38]. TLR4 enhances blastocyst attachment to endometrial cells in mice via miR-Let-7a suggesting that inflammatory responses are beneficial in the fetomaternal interface and supposedly accelerate the chances for successful implantation. In our study, TLR1 was markedly up expressed from oocyte to blastocyst development [39]. Therefore, the role of inflammatory responses is interesting and needs further study. GO and KEGG analyses of DEGs were then conducted. At different stages, GO analysis indicated that there was a significant over-representation of elements involved in nuclear speck and cytoplasm. The KEGG analysis revealed that most of the DEGs were primarily involved in the spliceosome, ribosome biogenesis in eukaryotes, and the Epstein–Barr virus infection pathway, RNA transport, ribosome pathway, transferase activity, and others. The above pathways have also been reported in the development of bovine and monkey embryos [11,40,41]. Therefore, the transcriptome profiles in sheep from oocyte to blastocyst development showed the same pattern. 

We then characterized the features of the lncRNAs and their target genes. The lncRNAs in cell samples from the oocyte, fertilized egg, 2-, 4-, 8-, 16-cell, morula, and blastocyst stages had lower expression levels compared with the expression levels observed in protein-coding genes. However, the transcript lengths of lncRNAs were mostly ≥1000 bp, and the mean transcript length of lncRNAs was not significantly different relative to the mean transcript length of protein-coding genes (Figure 5B). In addition, there was a total of 42 differentially expressed lncRNAs (52 transcript isoforms) (Figure 6). The major changes in lncRNA expression occurred between the morula and blastocyst stages, of which 19 (23 transcript isoforms) lncRNAs were significantly up-regulated, such as lnc MSTRG.2676, lnc MSTRG.3585, and six (10 transcript isoforms) lncRNAs were down-regulated, such as lnc MSTRG.8262 and lncMSTRG.11966 (Figure 6). Overall, there were fewer differentially expressed lncRNA transcripts during sheep development compared with the number of differentially expressed transcripts detected in implantation and inter-implantations sites in day-5 pregnant mice [30]. GO and KEGG analyses of the cis- and trans-target genes of the lncRNAs in the eight comparison groups were performed. GO analysis of the cis lncRNA targets revealed the following significantly enriched GO terms: RNA binding, nucleus, and others. The KEGG analysis revealed that the significantly enriched cis-pathways were “Huntington’s disease” and “Oxidative phosphorylation” between the morula and blastula groups (corrected *p*-value < 0.05, Figure 7G and Appendix A).

We then studied the distribution of different SNP and indel types (Table 4). Approximately 10% intergenic and 30% non-coding SNPs have been reported in humans from RNA-seq data [42]. The high percentages of intergenic SNPs in humans may be partially explained by the incomplete annotation of protein-coding genes and exons in the current version of the rainbow trout reference genome sequence [35]. We then conducted GO analysis of the associated genes. In the biological process’s category, SNP genes were associated with various cellular processes that were primarily involved in development-related mechanisms, including regulation of the MyD88-dependent toll-like receptor signaling pathway, regulation of metabolic and oxidation-reduction processes, and protein translation. In the molecular function category, SNP-containing genes were associated with binding phosphoprotein, nucleic acid, and actin. In addition, many genes were associated with transferase, motor, oxidoreductase, and structural molecule activities. In the cellular component category, many of the genes were associated with the cytoplasmic compartment, membranes, myosin complex, and extracellular region compartment (Appendix A), which is consistent with the findings of previous studies [31].

Finally, we also detected AS events. There are relatively few studies that have reported the numbers of AS events during development from the oocyte to the blastocyst. In this study, splicing events were comprehensively analyzed for 24 single cells based on relevant RNA-seq data. For example, there was a total of 101,835 AS events that were detected in 54,266 genes, comprising 7003 AE events detected in 2603 genes in the oocyte stage (Figure 8A). AS can control the transcriptional identity of the maternal transcriptome by the RNA-binding protein, which is essential for the development of fertilized-competent oocytes [43]. Therefore, our comprehensive analysis of AS suggests that AS plays an important role in sheep development. GO and KEGG analyses showed that there were 13,663 genes with one or more AS events at different stages during sheep development based on the UpSet plot (Figure 10A). The most significant biological process terms were “regulation of transcription, DNA templated,” “transport”, and “proteolysis”. The three most significant cellular component terms were “membrane,” “nucleus,” and “integral component of membrane”. The significant terms for molecular function were “ATP binding,” “nucleotide binding”, and “nucleic acid binding” (Figure 10B). In a previous study, AS has been shown to play an important role in the protein-coding genes of mouse oocytes and zygotes by RNA-binding protein, and the correct combination of exons through AS ensures that gene isoforms are expressed for the specific context in which they are required [44,45]. Furthermore, the “metabolism”, “genetic information processing” and “human diseases” correlated pathways were mostly genes involved in purine metabolism, RNA transport, and pathways in cancer (Figure 10C). In metabolomic analyses of fetal germ cells in mice on embryonic day (E)13.5 and E18.5, purine metabolism was involved in demonstrating sex- and developmental stage-dependent changes in these processes [46]. RNA transport in our study was mostly genes involved which also as one of the important AS events is frequent across developmental stages and tissues in embryonic day 8.5, 9.5. and 11.5 mouse embryos and placenta [47]. Furthermore, we detected 13,663 genes with one or more AS events, suggesting that AS may play an important role in sheep development. Furthermore, “ATP binding (1498 genes),” “nucleotide binding (1441 genes),” and “nucleic acid binding (1412 genes)” were the three most enriched categories, suggesting that alternative protein isoforms with distinct functions were expressed. Thus, defects in splicing control might be able to induce losses in function and severe phenotypes and thus require further study [48].

## 5. Conclusions

Here, we conducted a scRNA-seq survey of cells from sheep from the oocyte to the blastocyst developmental stages. We then carried out experiments to identify and functionally analyze the transcriptome profiles, long non-coding RNAs, single-nucleotide polymorphisms (SNPs), and alternative splicing (AS). We found between the oocyte and zygote groups significantly down-regulated genes and the second-largest change in gene expression occurred between the 8- and 16-cell stages. We used various methods to construct a profile to characterize cellular and molecular features and systematically analyze the related GO and KEGG profile of cells of all stages from the oocyte to the blastocyst. This large-scale, single-cell atlas provides key cellular information and will likely assist clinical studies in improving preimplantation genetic diagnosis.

## Figures and Tables

**Figure 1 genes-14-01145-f001:**
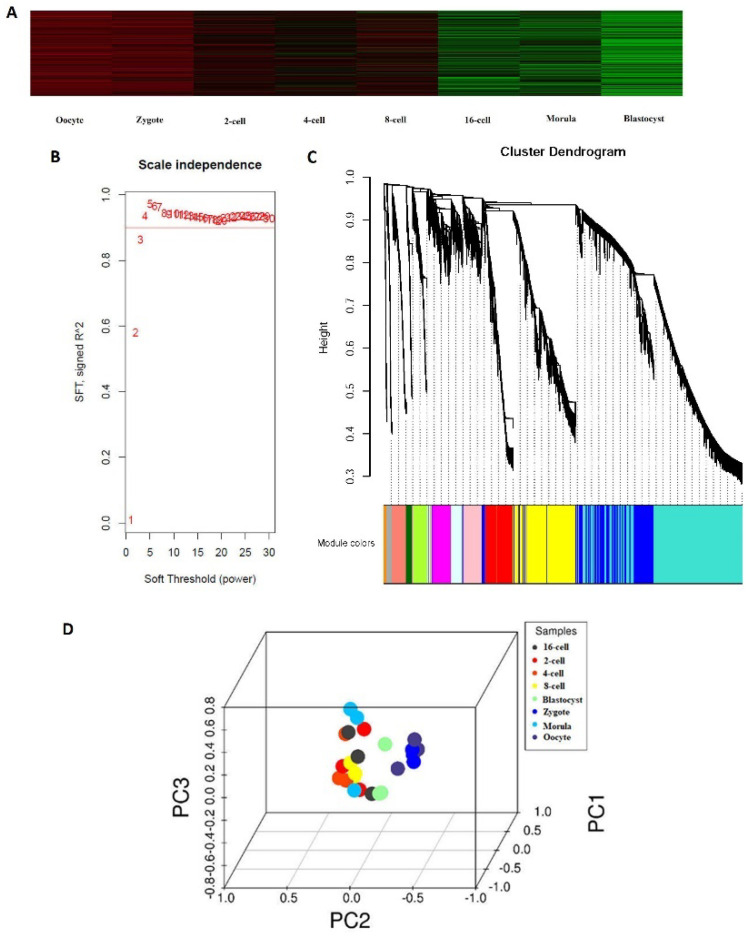
Expression profiles during sheep development from the oocyte to the preimplantation stage. (**A**) Heat map of gene expression profiles correlated with developmental stages. (**B**) Determination of the soft threshold with the WGCNA algorithm. The approximate scale-free fit index can be attained at the soft-thresholding power of 18. (**C**) Clustering dendrograms showing 26 modules containing a group of highly connected genes. Each color represents a certain gene module. (**D**) Principal component analysis. PC1, PC2, and PC3 represent the top three dimensions of the genes showing differential expression among preimplantation blastomeres.

**Figure 2 genes-14-01145-f002:**
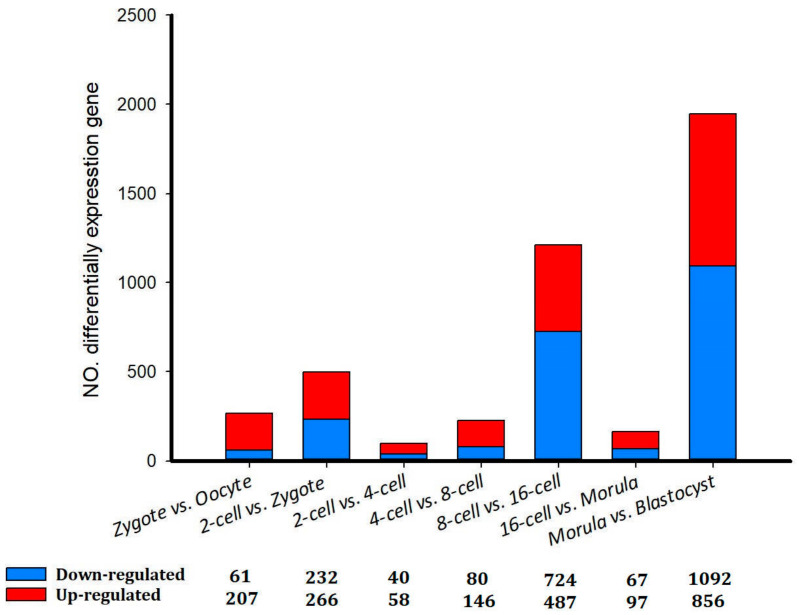
Differentially expressed genes in sheep development from the oocyte to the blastocyst stages.

**Figure 3 genes-14-01145-f003:**
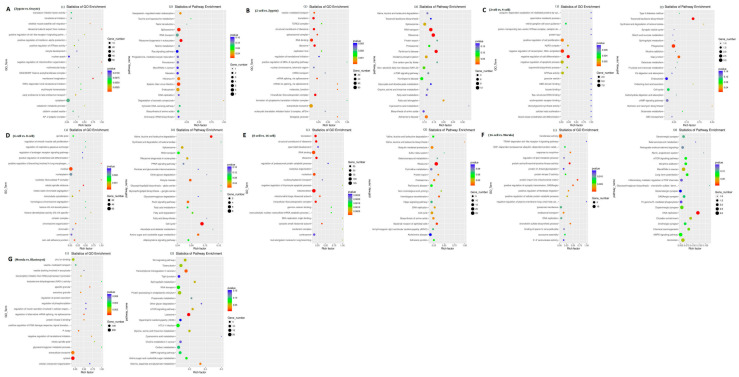
Gene ontology (GO) and Kyoto Encyclopedia of Genes and Genomes (KEGG) analyses of differentially expressed genes in sheep development from the oocyte to the blastocyst stages. GO and KEGG analyses between (**A**) the oocyte and zygote stages, (**B**) the zygote and 2-cell stages, (**C**) the 2-cell and 4-cell stages, (**D**) the 4-cell and 8-cell stages, (**E**) the 8-cell and 16-cell stages, (**F**) the 16-cell and morula stages, and (**G**) the morula and blastocyst stages.

**Figure 4 genes-14-01145-f004:**
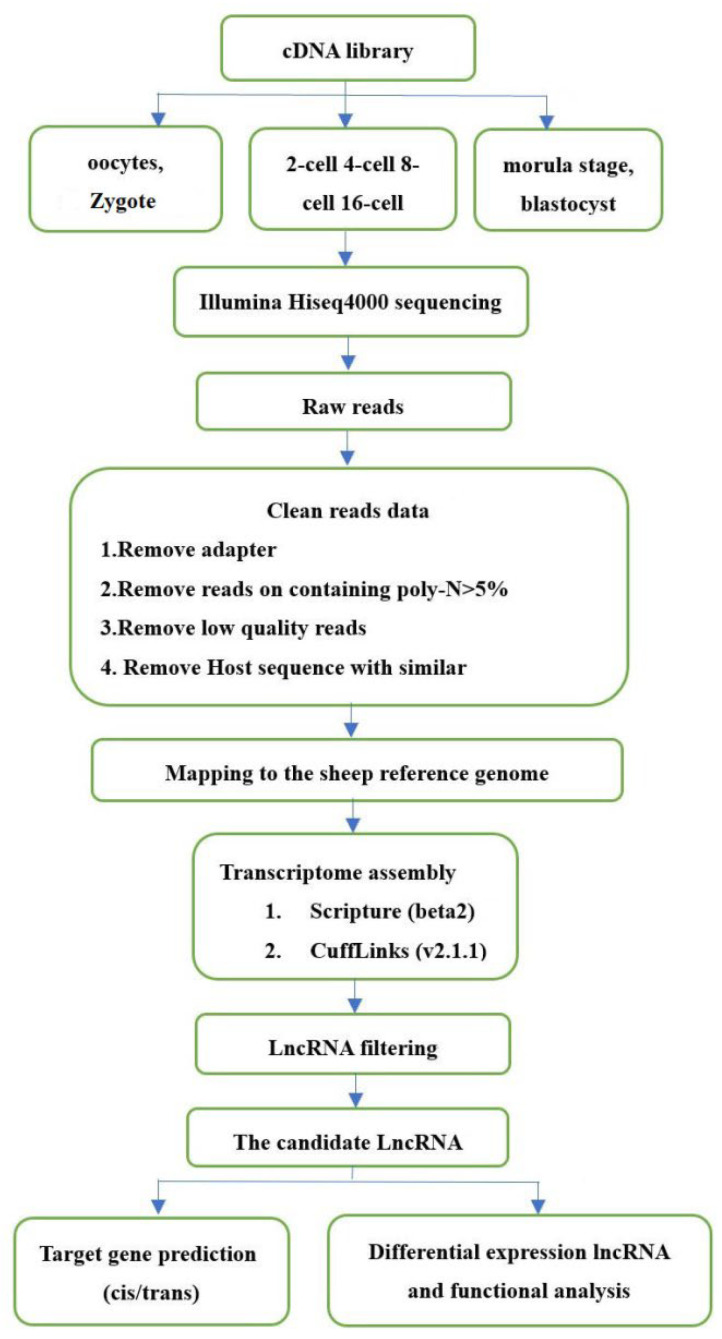
Pipeline for identification of lncRNAs.

**Figure 5 genes-14-01145-f005:**
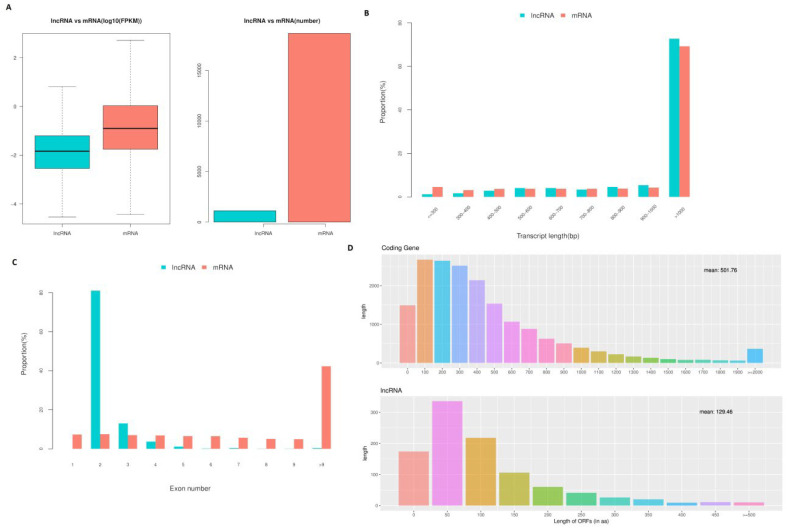
LncRNA characteristics in sheep development from the oocyte to the blastocyst stages. (**A**) Comparison of the expression level between lncRNA and protein-coding genes in terms of fragments per kilobase of exon per million fragments mapped (FPKM). The FPKM distribution of lncRNAs in mouse uterus was lower than that of protein-coding genes. (**B**) Distribution of transcript lengths in the lncRNAs and protein-coding genes. The average size of lncRNA transcripts was generally shorter than that of protein-coding genes. (**C**) The number of exons in lncRNAs and protein-coding genes. A total of 88.41% of the lncRNAs contained two to four exons, while the majority of protein-coding genes contained more than 10 exons. (**D**) The number of ORFs identified in the lncRNAs and protein-coding genes using Estscan. As expected, the ORFs of lncRNAs were substantially shorter than the ORFs in protein-coding genes.

**Figure 6 genes-14-01145-f006:**
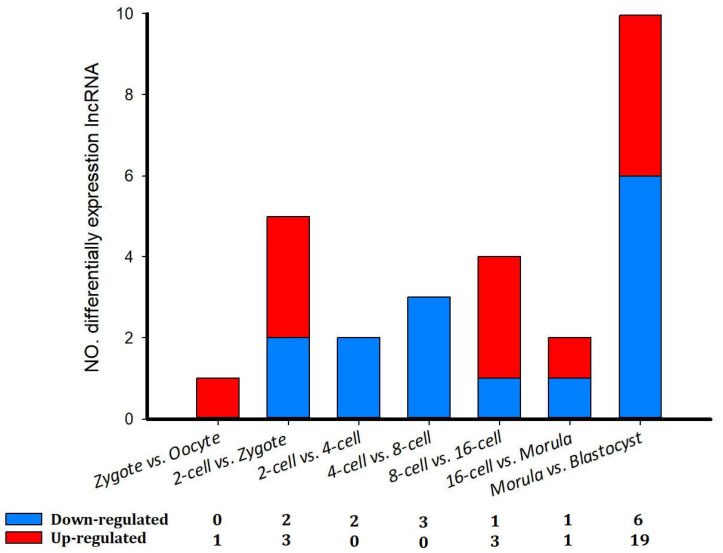
The number of differentially expressed lncRNAs in eight comparison groups in sheep development from the oocyte to the blastocyst stages.

**Figure 7 genes-14-01145-f007:**
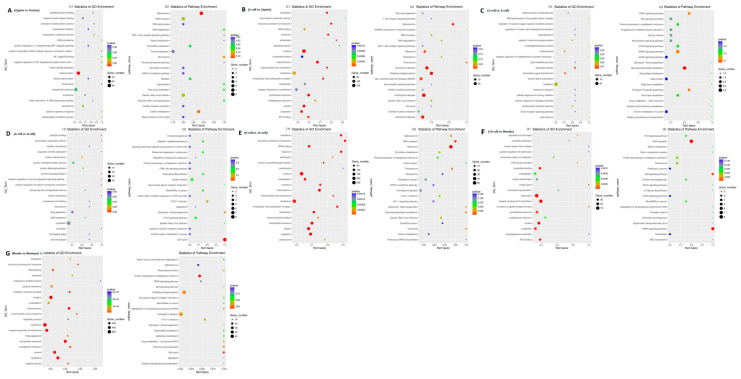
GO and KEGG analysis with the cis and trans-target genes of lncRNAs. GO and KEGG analyses between (**A**) oocyte and zygote stages, (**B**) zygote and 2-cell stages, (**C**) 2-cell and 4-cell stags, (**D**) 4-cell and 8-cell stages, (**E**) 8-cell and 16-cell stages, (**F**) 16-cell and morula stages, and (**G**) morula and blastocyst stages.

**Figure 8 genes-14-01145-f008:**
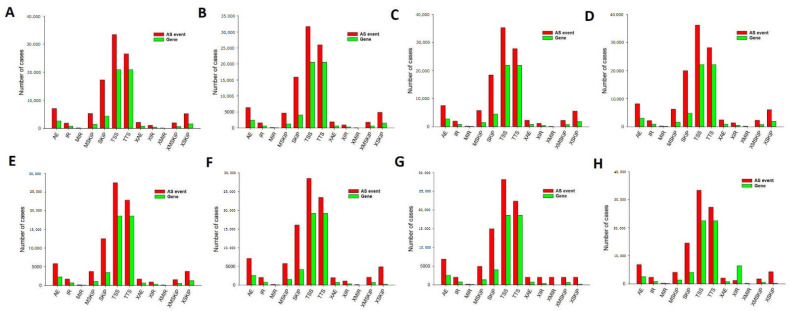
The number of alternative splicing events and associated genes in sheep development from the oocyte to the blastocyst stages. TSS was the most frequent of the eleven types of events. (**A**) Oocyte stage. (**B**) Zygote stage. (**C**) 2-Cell stage. (**D**) 4-Cell stage. (**E**) 8-Cell stage. (**F**) 16-Cell stage. (**G**) Morula stage. (**H**) Blastocyst stage. *AE* alternative exon ends, *AD* alternate donor, *AP* alternate promoter, *AT* alternate terminator, *ES* exon skip, *ME* mutually exclusive exon, *RI* retained intron.

**Figure 9 genes-14-01145-f009:**
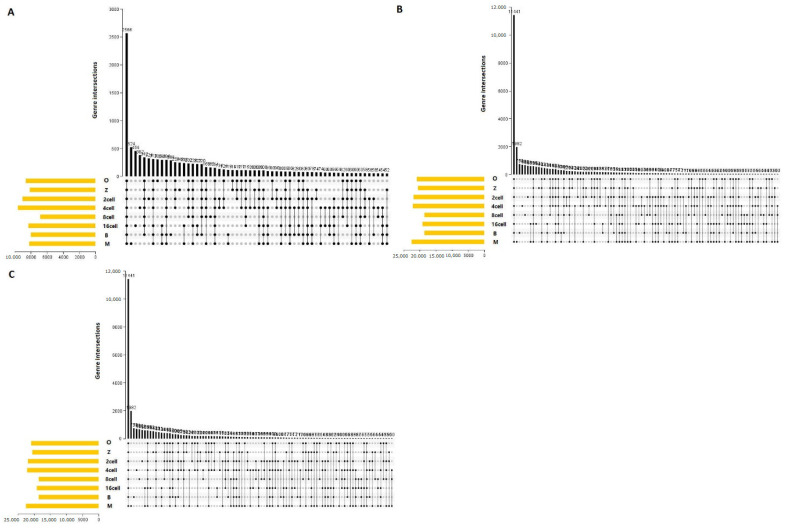
UpSet plot of the three main alternative splicing events at different stages in sheep development from the oocyte to the blastocyst stages. UpSet plot of the interactions of the alternative splicing events associated with genes. (**A**) UpSet plot of the interactions of SKIP. (**B**) UpSet plot of the interactions of TSS. (**C**) UpSet plot of the interactions of TTS. *TTS* alternative transcription termination site.

**Figure 10 genes-14-01145-f010:**
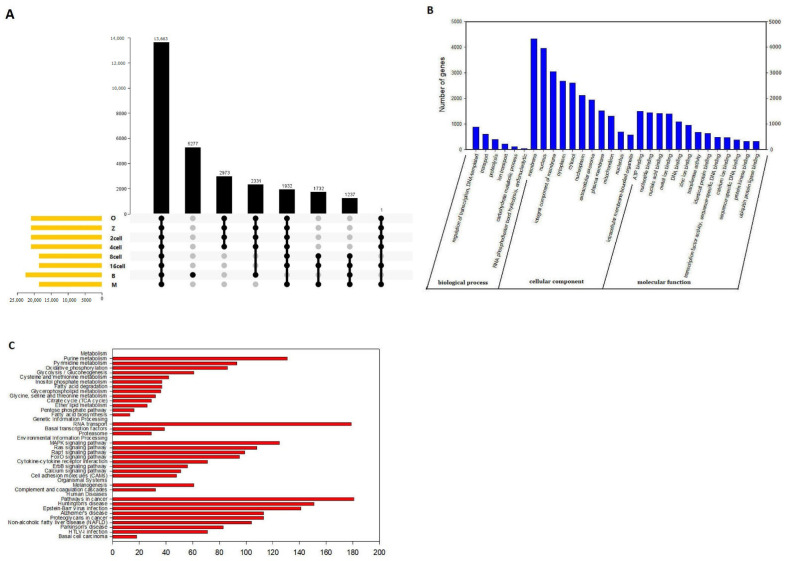
Molecular characteristics of the main alternative splicing events in sheep development from the oocyte to the blastocyst stages. (**A**) UpSet plot of the main genes at different stages in sheep development from the oocyte to the blastocyst stages. (**B**) GO analysis. (**C**) KEGG analysis.

**Table 1 genes-14-01145-t001:** Numbers of embryos and cells analyzed by single-cell RNA-Seq analysis.

Sample	No. of Sample	No. of Single Cells
Oocyte	3	3
Zygote	3	3
2-cell	3	3
4-cell	3	3
8-cell	3	3
16-cell	3	3
Morula	3	3
Blastocyst	3	3
Total	24	24

**Table 2 genes-14-01145-t002:** The numbers of genes detected in sheep from oocyte to blastocyst development.

Stage	No. of Genes (FPKM > 0.1)
Oocyte	24,729
Zygote	25,455
2-cell	26,714
4-cell	28,948
8-cell	25,920
16-cell	24,341
Morula	22,242
Blastocyst	31,157

FPKM: fragments per kilobase of exon per million fragments mapped.

**Table 3 genes-14-01145-t003:** Summary of read filter and alignment.

Sample	Raw Reads	Clean Reads	Clean Bases	Error Rate (%)	GC Content (%)
oocyte	102,383,290	86,741,418	13.01 G	0.04	43.50
Zygote	111,921,426	87,081,144	13.06 G	0.02	44.50
2-cell	123,273,652	95,679,634	14.35 G	0.04	44
4-cell	108,522,186	96,419,102	14.46 G	0.03	44
8-cell	142,178,172	123,561,104	18.53 G	0.02	45
16-cell	133,023,048	90,477,914	13.57 G	0.04	43.5
morula	122,556,646	90,506,796	13.58 G	0.02	44.5
blastula	141,488,370	105,354,406	15.80 G	0.03	44

**Table 4 genes-14-01145-t004:** Summary of SNP and indel classification for different sets.

Functional Class	SNP								INDEL							
	Oocyte	Zygote	2-Cell	4-Cell	8-Cell	16-Cell	Morula	Blastocyst	Oocyte	Zygote	2-Cell	4-Cell	8-Cell	16-Cell	Morula	Blastocyst
Intergenic	79,516	70,858	99,129	115,926	72,615	78,782	66,420	224,557	5650	5260	7088	8129	6163	4939	4765	15,130
Intronic	47,852	40,178	60,970	67,126	37,762	64,925	39,054	201,642	3104	2831	3973	4172	2793	3580	2566	15,373
exonic	28,898	24,940	33,951	35,201	26,715	32,414	27,569	66,325	336	298	438	411	415	326	440	512
3′UTR	3991	4763	5879	5761	4773	4822	3991	8356	502	564	666	651	598	547	577	862
5′UTR	1075	904	1142	1269	964	1265	1075	2145	34	32	35	42	35	41	44	60
upstream	4343	3612	5330	5836	3937	5551	4343	9568	243	233	359	408	285	306	299	588
downstream	14,572	16,348	19,799	20,220	16,904	16,251	14,572	27,967	1845	1814	2268	2209	2072	1749	2010	2886
splicing	632	375	421	494	390	390	632	525	47	43	53	52	47	44	53	71
Total number	184,380	162,027	226,690	251,891	164,117	204,485	157,743	541,244	11,763	11,076	14,882	16,076	12,408	11,534	10,755	35,487

## Data Availability

Not applicable.

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
