# Peer review of "Identification and Functional Analysis of Transcriptome Profiles, Long Non-Coding RNAs, Single-Nucleotide Polymorphisms, and Alternative Splicing from the Oocyte to the Preimplantation Stage of Sheep by Single-Cell RNA Sequencing"

_genes, 2023, doi:10.3390/genes14061145_

Round 1
Reviewer 1 Report
The manuscript is well-edited and well-written, presenting a huge body of new research. I suggest a few changes in a few places as follows:
Line 32-34. “We then carried out experiments to identify and functionally analyze the transcriptome profiles, long non-coding RNAs, single-nucleotide polymorphisms (SNPs), and alternative splicing (AS).” It is a repetition of line 28-30. So, I propose to delete or rephrase.
Line 34-35: “We founded that between 34 the oocyte and zygote groups significantly down-regulated genes …”. I propose to edit the word order of the first part of the sentence by omitting "that".
Line 42: “Single-cell” is to write with lowercase initial.
Line 86: “lnc RNAs”, delete space.
Line 82: “(ESCs)”, correct typing style.
Line 98: “dose compensation”, do you mean dose compensation or dosage compensation?
Line 106: Please include the full names of abbreviations (GO and KEGG) in brackets.
Line 114: I suggest capitalizing the initial letter of several proper nouns.
Line 118: “bulls”. I suggest replacing it with a ram or a sire. The sheep's reproduction is strongly influenced by many environmental factors, and it is worth briefly discussing these in the context of this study.
Line 119-122: “Donor animals were sacrificed at 30 hours” and “Early morulae, compact morulae, and blastocysts were collected by routine non-surgical uterine flushing on days 5, 6, and 7.” The description of the experiment should be clarified. The authors should explain what sacrifice means. The two sentences quoted are contradictory. Resolve the contradiction. Give reasons why IVF has not been used.
Line 133: “Approximately 10 ug of total RNA …”. Do you mean ug or μg?
Line 173: Please include the full name of abbreviation (FPKM) in bracket.
Line 253: The principal component analysis (PCA) is missing from that section.
Line 258: “24 cells”. For better clarity, I propose: 24 cell samples.
Figure 2.: I propose to rename the first two columns of the figure according to the logic of the pairing used later.
Line 260 and 312: According to the guide, standardize references to tables and figures here and everywhere! Also, in many places, put a space before the new sentence because it is missing.
Table 3. and Figure 4.: Why has the word zygote been changed to fertilized egg?
Line 433: For a more accurate comparison between sheep and mice, it is necessary to specify the day associated with each embryonic stage. In the sheep, implantation is typical after the second week.
Figure 6: see remark at Figure 2.
Line 437: cis is to write with capital initial.
Author Response
Response to comments of reviewers: Reviewer #1 Thank you for your careful review and constructive comments. Question: 1. Line 32-34. “We then carried out experiments to identify and functionally analyze the transcriptome profiles, long non-coding RNAs, single-nucleotide polymorphisms (SNPs), and alternative splicing (AS).” It is a repetition of line 28-30. So, I propose to delete or rephrase. Response:Thanks for your advice. We rephrased this sentence and marked red. Question: 2. Line 34-35: “We founded that between 34 the oocyte and zygote groups significantly down-regulated genes …”. I propose to edit the word order of the first part of the sentence by omitting "that". Line 42: “Single-cell” is to write with lowercase initial. Line 86: “lnc RNAs”, delete space. Line 82: “(ESCs)”, correct typing style. Response:Thanks for your advice. We are sorry for our careless. And then, we rephrased this sentence and marked red. Question: 3.Line 98: “dose compensation”, do you mean dose compensation or dosage compensation? Line 106: Please include the full names of abbreviations (GO and KEGG) in brackets. Line 114: I suggest capitalizing the initial letter of several proper nouns. Line 118: “bulls”. I suggest replacing it with a ram or a sire. The sheep's reproduction is strongly influenced by many environmental factors, and it is worth briefly discussing these in the context of this study. Response:Thanks for your advice. We author agree with it. dose compensation changed as dosage compensation. GO (Gene Ontology) and KEGG (Kyoto Encyclopedia of Genes and Genomes), Henan academy of agricultural sciences institutional animal care and use committee with the capitalizing the initial letter. Bulls changes as rams. And then, we rephrased this sentence and marked red. Question: 4. Line 119-122: “Donor animals were sacrificed at 30 hours” and “Early morulae, compact morulae, and blastocysts were collected by routine non-surgical uterine flushing on days 5, 6, and 7.” The description of the experiment should be clarified. The authors should explain what sacrifice means. The two sentences quoted are contradictory. Resolve the contradiction. Give reasons why IVF has not been used. Line 133: “Approximately 10 ug of total RNA …”. Do you mean ug or μg? Line 173: Please include the full name of abbreviation (FPKM) in bracket. Line 253: The principal component analysis (PCA) is missing from that section. Line 258: “24 cells”. For better clarity, I propose: 24 cell samples. Response:Thanks for your advice. We rephrased this sentence and marked red. Sacrificed is wrong and corrected as to anesthetize. μg is correct! FPKM (Fragments Per Kilobase Million). We changed 24 cells as 24 cell samples.Thank you so much! Question: 4 Figure 2.: I propose to rename the first two columns of the figure according to the logic of the pairing used later. Line 260 and 312: According to the guide, standardize references to tables and figures here and everywhere! Also, in many places, put a space before the new sentence because it is missing. Table 3. and Figure 4.: Why has the word zygote been changed to fertilized egg? Line 433: For a more accurate comparison between sheep and mice, it is necessary to specify the day associated with each embryonic stage. In the sheep, implantation is typical after the second week. Figure 6: see remark at Figure 2. Line 437: cis is to write with capital initial. Response:Thanks for your advice. We rephrased this sentence and marked red. the word zygote been changed to fertilized egg for our careless! We changed at the Table 3. and Figure 4 as zygote.Line 433, yes, we agree with the reviewer and deleted the time. Figure 6 is shown the number of LncRNA,but Figure 2 is the Differentially expressed genes.
Reviewer 2 Report
REVIEW
for the journal Genes (ISSN 2073-4425)
Article
“Identification and functional analysis of transcriptome profiles, long non-coding RNAs, single-nucleotide polymorphisms, and alternative splicing from the oocyte to the preimplantation stage of sheep by single-cell RNA sequencing”
Manuscript ID: genes-2354463
Authors: Zi Jing Zhang, Qiao ting Shi, Xiao ting Zhu, Lei Jin, Li min Lang, Shi jie Lyv, Xiao ling Xin, Yong zhen Huang, Xiang Yu, Zhi ming Li, Su Juan Chen, Zhao xue Xu, Wei Zhang, Er yao Wang
The authors performed a) scRNA-seq study of 24 sheep cells during their development (from the oocyte stage to the blastocyst stage); b) experiments involved in the identification and functional analysis of transcriptome profiles, lncRNAs, single-nucleotide polymorphisms (SNPs) and alternative splicing (AS). They have created a profile that characterizes cellular and molecular properties, and this large-scale single-cell atlas could help develop more efficient preimplantation genetic diagnosis in clinical trials.
I would like to make a few comments that might improve the quality of this interesting article.
1) The abbreviations GO (Gene ontology) and KEGG (Kyoto Encyclopedia of Genes and Genomes) are explained by the authors only in text lines 339-340. Recommendation: the explanation of abbreviations should be provided in the text the first time they are used.
2) Citation numbers of literary sources in the text should not be indicated in the index.
3) Line 175. Authors should indicate which version of the R package was used for flexible differential expression analysis of RNA-Seq data (Ballgown).
4) The statistical analysis section (lines 247 - 253) should be detailed considering the purpose and design of the study. For example, line 304 indicates that the authors used principal component analysis. This and other statistical methods used should be specified in the article's methodology.
5) The axis information in Figure 1 is blurry and hard to read.
6) The information in Figure 3 is hard to read.
7) The same remark applies to Figures 5, 7, 8, 9 and 10.
8) The title of section 3.7 should start with a capital letter.
9) Authors should review the reference list to ensure that the bibliographic descriptions meet the requirements of the journal. For example, the year must be highlighted.
10) The study described in the article is very extensive and I have no doubt about the relevance of the article, theoretical and practical benefits, systematic analysis, representativeness of the data and the consistency of the conclusions, but I recommend the authors to make these corrections.
Sincerely, reviewer.
Author Response
Response to comments of reviewers: Reviewer #2 Thank you for your careful review and constructive comments. Question: 1) The abbreviations GO (Gene ontology) and KEGG (Kyoto Encyclopedia of Genes and Genomes) are explained by the authors only in text lines 339-340. Recommendation: the explanation of abbreviations should be provided in the text the first time they are used. 2) Citation numbers of literary sources in the text should not be indicated in the index. Response:Thanks for your advice. We rephrased this sentence and marked red. Question:3) Line 175. Authors should indicate which version of the R package was used for flexible differential expression analysis of RNA-Seq data (Ballgown). 4) The statistical analysis section (lines 247 - 253) should be detailed considering the purpose and design of the study. For example, line 304 indicates that the authors used principal component analysis. This and other statistical methods used should be specified in the article's methodology. 5) The axis information in Figure 1 is blurry and hard to read. 6) The information in Figure 3 is hard to read. 7) The same remark applies to Figures 5, 7, 8, 9 and 10. Response:Thanks for your advice. We rephrased this sentence and marked red. First,we used R package (1.12.0) for flexible differential expression analysis of RNA-Seq data. Second, we supplied the PCA in the part of 2.5.And then, we also supplied the new fig1 and fig3 and 5, 7, 8, 9 and 10 would be sublimed to the editor. Thank you so much! Question:8) The title of section 3.7 should start with a capital letter. 9) Authors should review the reference list to ensure that the bibliographic descriptions meet the requirements of the journal. For example, the year must be highlighted. 10) The study described in the article is very extensive and I have no doubt about the relevance of the article, theoretical and practical benefits, systematic analysis, representativeness of the data and the consistency of the conclusions, but I recommend the authors to make these corrections. Response:Thanks for your advice. We rephrased this sentence and Thank you so much!.
